# Evaluation of the Determination of Dabigatran, Rivaroxaban, and Apixaban in Lupus Anticoagulant-Positive Patients

**DOI:** 10.3390/diagnostics11112027

**Published:** 2021-11-02

**Authors:** Jana Úlehlová, Barbora Piskláková, Eliška Ivanovová, Jana Procházková, Pavla Bradáčová, Aleš Kvasnička, David Friedecký, Luděk Slavík

**Affiliations:** 1Department of Hematology-Oncology, Faculty of Medicine and Dentistry, Palacký University and University Hospital Olomouc, 779 00 Olomouc, Czech Republic; jana.ulehlova@fnol.cz (J.Ú.); jana.prochazkova@fnol.cz (J.P.); pavla.bradacova@kzcz.eu (P.B.); 2Laboratory for Inherited Metabolic Disorders, Department of Clinical Biochemistry, Palacký University and University Hospital Olomouc, 779 00 Olomouc, Czech Republic; barbora.pisklakova@upol.cz (B.P.); eliska.ivanovova@gmail.com (E.I.); ales.kvasnicka01@upol.cz (A.K.)

**Keywords:** dabigatran, rivaroxaban, apixaban, DOAC-Stop, liquid chromatography tandem mass spectrometry

## Abstract

Background: The effect of direct oral anticoagulants (DOAC) on laboratory tests dependent on the production of their targets, factor IIa and factor Xa, is a well-known problem and can cause both false positive and negative results. In particular, the situation in patients who develop lupus anticoagulant (LA) antibodies is highly complex. To evaluate the effectiveness of DOAC therapy in lupus-positive patients, 31 samples were enrolled in this retrospective study. All patient samples were spiked with three types of DOAC (dabigatran, DABI; rivaroxaban, RIVA; and apixaban, API) in a concentration that significantly influenced the screening test for LA and thus can mask the presence of LA. Subsequently, the DOAC was always unbound by the DOAC-Stop procedure. DOAC levels before and after binding were determined by functional assays, followed by liquid chromatography coupled with mass spectrometry (LC-MS) analysis. Methods: The determination of DOAC levels was performed by direct thrombin assay and determination of anti-Xa activity with specific calibration as functional tests for DABI and xabans (API and RIVA). To determine concentration levels of API, DABI, and RIVA, our in-house LC-MS method was used. Results: The results of LA-positive samples show significant differences between functional tests and the LC-MS method both before and after DOAC binding. Conclusions: The acute findings of the presence of LA-type antibodies fundamentally affects the determination of DOAC by functional tests, and in this case, it is necessary to use LC-MS analysis to determine the true value. If patients treated with DOAC develop LA of medium and higher titers, we do not recommend checking DOAC levels with functional tests.

## 1. Introduction

Recently, several publications have addressed the impact of lupus anticoagulant (LA) testing in patients treated with new direct oral anticoagulants (DOAC), who have progressively developed lupus anticoagulant antibodies. In this study, however, we ask the opposite question. Is the determination of DOAC levels in these patients correct? Is it not affected by the presence of antibodies that interfere with the functional tests for determining DOAC levels?

Although warfarin is still the preferred treatment for patients with LA (except for APS syndrome) due to its extensive clinical experience, many patients are currently treated primarily with DOAC in practice. Furthermore, patients may also develop LA secondary to the course of treatment for atrial fibrillation, the prevention of ischemic stroke and thromboembolic events. These patients, in whom DOAC levels need to be monitored in some situations, are the focus of our study.

Until now, all work has proven DOAC control by specific tests to be highly reliable [1]. Liquid chromatography with tandem mass spectrometry (LC-MS) is the definitive tool for measuring drugs and their metabolites, and it is the predominant method used in the development and testing of DABI [2]. This technique is not affected by changes in clotting factor levels, the presence of LA, or preanalytical variables that can affect many clotting tests. However, it has limited availability, it is time consuming and expensive, and requires a high level of technical knowledge. Thus, LC-MS is not a practical choice for most diagnostic laboratories, nor is it suitable for rapid time requirements in clinical care.

Commercially available diluted thrombin time (dTT), the Hemoclot^®^ Thrombin Inhibitor (HTI) test (Hyphen BioMed, Neuville-sur-Oise, France), is a simple, fast, sensitive, and quantitative method to measure the concentrations of direct thrombin inhibitors in the blood. The dilution of the test plasma exceeds the extreme sensitivity to DABI observed with thrombin time (TT). The diluted test plasma is mixed with normal plasma and a constant amount of purified human α-thrombin is added to initiate coagulation. The test shows a linearity between DABI concentration and coagulation time with good reproducibility, and is easy to use for automation [3,4].

Anti-Xa chromogenic assays can accurately measure a wide range of plasma concentrations, provided that a standard calibration curve is established with calibrators and controls of RIVA or API calibrators and controls [5,6]. A study of 23 centers showed that chromogenic assays against factor Xa in conjunction with RIVA calibrators and controls can measure plasma concentrations of RIVA in the range of 20–660 μg/L, but limited data are described for API [6]. In addition, the mean measured concentrations of RIVA were in agreement with the expected values, even at low RIVA concentrations, when a modified STA^®^ Rotachrom^®^ test (Diagnostica Stago) was used.

## 2. Materials and Methods

### 2.1. Patients

This retrospective study was carried out on a set of blood samples sent to our laboratory from 31 patients with LA diagnosed according to the recommendation of the International Society on Thrombosis and Haemostasis [7]. Patients were not treated with any anticoagulant drugs. Blood samples containing LA were spiked with DABI, RIVA, and API. Calibrators (100 uL) with specific DOAC were added to platelet-poor plasma from citrated blood (0.5 mL), which corresponds to the upper limit of recommended dose of DOAC. Citrated plasma samples were stored at −80 °C before analysis.

### 2.2. Blood Sampling

Blood (3.5 mL) was collected with a Vacuette^®^ needle (Greiner Bio-One, Vienna, Austria) into a vacuum tube with a buffered solution containing sodium citrate at a concentration of 0.109 mol/L (3.2%). The system ensured a blood and anticoagulant mixture in the desired 1:10 ratio. The blood was then carefully mixed in a test tube, with the tube gently turned upside down several times and transported to the laboratory. Next, the sample was centrifuged for 10 min at 3000× *g*, after which 0.5 mL of the upper layer of platelet-poor plasma was aspirated, frozen and stored at −80 °C until analysis was performed. In the analysis of the LA, the sample was repeatedly centrifuged under the same conditions. For the actual analysis, the sample was thawed at room temperature.

### 2.3. The DOAC-Stop Procedure

DOAC levels were determined before and after the addition of adsorbent tablets (DOAC-Stop; Haematex Research, Hornsby, Australia), according to the manufacturer’s instructions and depending on the plasma volume. Briefly, a half-tablet of DOAC-Stop designed to adsorb DOAC was added to each 0.5 mL of plasma. Subsequently, the sample was gently mixed for 5 min and centrifuged for 2 min at 3000× *g* to precipitate the DOAC with adsorbent. Finally, the supernatant, which is conjectured to be free of DOAC, was collected to be further analyzed. The composition of DOAC-Stop is proprietary information. The concentrations of API, DABI, and RIVA were also measured before and after the DOAC-Stop procedure.

### 2.4. Diluted Thrombin Time

The assay for specific measurements of DABI with highly purified human α-thrombin was performed using the ACL TOP^®^ 700 CTS system, the HemoClot^®^ Thrombin Inhibitors kit (HYPHEN BioMed, Neuville-sur-Oise, France), and the HemosIL^®^ Direct Thrombin Inhibitor assay (IL, Lexington, KY, USA). The reference range for the dTT assay is not defined and depends on the strength of the anticoagulant treatment. Both tests are designed for clinical use and affixed with the CE mark.

### 2.5. Anti-Xa Activity Assay

The assay for specific measurements of RIVA and API was performed using the ACL TOP^®^ 700 CTS system and the HemosIL^®^ Heparin kit (IL, Lexington, USA), and calibrated for RIVA with the Technoview^®^ Rivaroxaban set (Technoclone, Vienna, Austria) and for API with Technoview^®^ Apixaban (Technoclone, Vienna, Austria). The reference range for anti-Xa is not defined and depends on the strength of the anticoagulant treatment. The test is designed for clinical use and is affixed with the CE mark.

### 2.6. Liquid Chromatography Tandem Mass Spectrometry (LC-MS)

LC-MS analysis was performed using the liquid chromatography system (Dionex, Sunnyvale, CA, USA) coupled with a triple quadrupole 6500 tandem mass spectrometer (AB Sciex, Foster City, CA, USA) according to the previously published method [8,9]. Dabigatran, rivaroxaban, and stable isotopically labelled D3-dabigatran (Toronto Research Chemicals Inc, Toronto, ON, Canada) and apixaban (Pfizer Inc., New York, NY, USA) were dissolved in methanol (LC-MS quality, Sigma, St. Louis, MO, USA) at a final concentration of 1 mg/mL expressed as free substances. These stock solutions were used for the preparation of all other standards. For quantification, a series of calibration standards in methanol were prepared (concentrations 0, 10, 50, 100, and 500 ng/mL of DABI, API, and RIVA). The calibration standards were prepared in addition to drug-free plasma from healthy volunteers. All solutions were stored at −20 °C.

### 2.7. Lupus Anticoagulants

LA tests were performed using LAC Screen tests (Werfen, Barcelona, Spain) with improved specificity based on the diluted Russell’s viper venom tests (dRVVT). The test detects the group of antiphospholipid antibodies that are directed against negatively charged phospholipids or complexes between phospholipids and proteins (either beta-2-glycoprotein 1 or clotting factors such as prothrombin) and prolonged coagulation tests. The dRVVT test is performed as a screening with low phospholipid concentration and confirmation with high phospholipid concentration, and is always expressed as a ratio to normal plasma. Pathology is defined by a ratio greater than 1.2.

For the purpose of evaluation of the effect of DOAC on dRVVT assays, we performed measurements using standard plasma spiked with a known amount of DOAC from 0 to 250 ng/mL to evaluate the effect of DOAC itself on dRVVT.

### 2.8. Statistical Analysis

Statistical analysis was performed using the software GraphPad Prism 9.0 (GraphPad Software, San Diego, CA, USA). Normality of the data was tested, and according to its results nonparametric tests were used. Spearman correlation analysis was used for comparison of functional assays (anti-Xa and dTT) with LC-MS as a reference method. Differences between before and after the DOAC procedure were evaluated using the nonparametric paired Wilcoxon test and the unpaired Mann‐Whitney test. The level of statistical significance for all tests was set at 95%. Probability values of *p* < 0.05 were considered statistically significant.

## 3. Results

In our study, 31 samples of patients with the presence of potent LA with different DOAC were examined. The strength of the inhibitor was determined by the dRVVT test.

### 3.1. The dTT and Anti-Xa Tests Do Not Correlate with LC-MS in Samples from Patients with Lupus

The results of individual direct anticoagulant levels were evaluated by each method, functional assays and LC-MS for individual anticoagulants (left part of Figure 1) and after the DOAC-Stop procedure, which removed 99% of the direct anticoagulant.

The results show that the samples from LA patients show significant variation in DOAC levels by functional assays and LC-MS both after DOAC addition and removal (Appendix A). Descriptive statistics including Shapiro‐Wilk normality testing and data distribution visualized by violin plots were performed (Table 1). Except for one group (DABI_LC-MS), all groups did not show normal distribution of data, and therefore non-parametric tests were applied for further statistical evaluation.

Statistical evaluation shows a significant difference between the two methods, which is due to interference of the functional assays. When the results before and after the DOAC procedure are compared, the ratios of medians for the functional tests are in the range of 1.4–1.7, whereas for LC-MS in the range of 62–183. All comparisons were evaluated as statistically significant with a *p*-value < 0.05. Furthermore, the median ratios of dTT and Anti-Xa compared with LC-MS before the DOAC procedure were calculated as 1.9, 1.1 and 0.5 for DABI, RIVA and API, respectively.

### 3.2. dTT and Anti-Xa Strongly Correlate with LA Ratio

Strong correlations were observed between dTT / Anti-Xa and LA ratio, whereas uncorrelated LC-MS results were observed. Based on the experiment design, the values of Spearman’s correlation coefficient r > 0.36 were considered statistically significant [10]. The results of the data analysis (correlation coefficients and *p*-values) are shown in Table 2 and Figure 2.

## 4. Discussion

The new oral anticoagulants are revolutionary drugs in the field of anticoagulation and show clearly better results in antithrombotic prevention after the era of Vitamin K antagonists. Although their main advantage is the use of fixed rather than individual doses in each therapeutic or preventive indication, there are situations requiring the monitoring of drug levels.

Since the beginning of the use of DOAC, their presence has been described to affect coagulation tests [11,12]. Several publications have established their influence on individual tests ranging from determination of factor levels to thrombophilia [13,14,15]. The greatest influence has been observed in the determination of LA, with up to half of the samples being misdiagnosed in the presence of LA [16]. For these cases, functional screening coagulation methods (prothrombin time and activated partial thromboplastin time) were used but did not show sufficient specificity for DOAC [17]. Therefore, specific functional assays have been developed for DABI, RAVI and API based on dTT [18] and the anti-Xa activity, respectively. These tests showed significantly better specificity to DOACs [19], with the exception of the administration of heparin. Unfractionated heparin, whose main target is thrombin, affects dTT, whereas low molecular weight heparin influences the specific determination of factor-Xa activity, where it is the main target. Moreover, some papers have reported possible interferences with these assays especially in the context of preanalytical variability of the samples (lipemia, bilirubin, and haemolysis) [20].

Based on results, preanalytical conditions as well as the presence of some coagulation disorders such as LA may cause interference in otherwise robust specific functional assays for DOAC, whether based on dTT or anti-Xa activity.

## 5. Conclusions

The results show that not only treatment with new anticoagulants can affect many coagulation tests, but also reverse coagulation changes in a given patient can interfere with the determination of DOAC levels. In our case, we confirmed the presence of LA, i.e., antibodies containing phospholipid surfaces necessary for activation reactions in functional coagulation assays that are used to determine DOAC levels. Our work using a comparison of functional assays with LC-MS has allowed us to determine the degree of interference and shows that although functional specific assays have significantly better specificity than screening coagulation assays in the determination of DOAC, there are situations where LC-MS should be used.

## Figures and Tables

**Figure 1 diagnostics-11-02027-f001:**
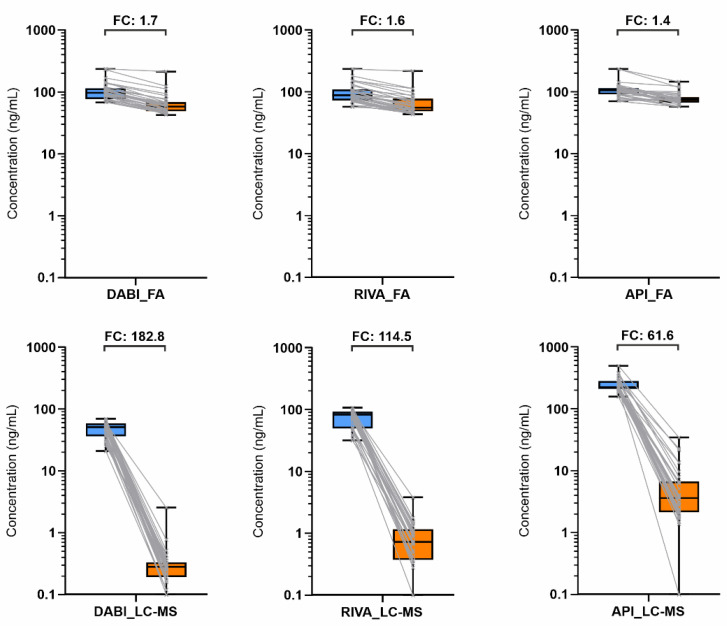
Distribution of DABI, RIVA, and API levels before (left blue) and after (right orange) DOAC procedure performed by dTT/anti-Xa (functional assays, FA) and LC-MS with the aforementioned fold changes (FC). Abbreviations: DABI, dabigatran; RIVA, rivaroxaban; API, apixaban; FA, functional assay.

**Figure 2 diagnostics-11-02027-f002:**
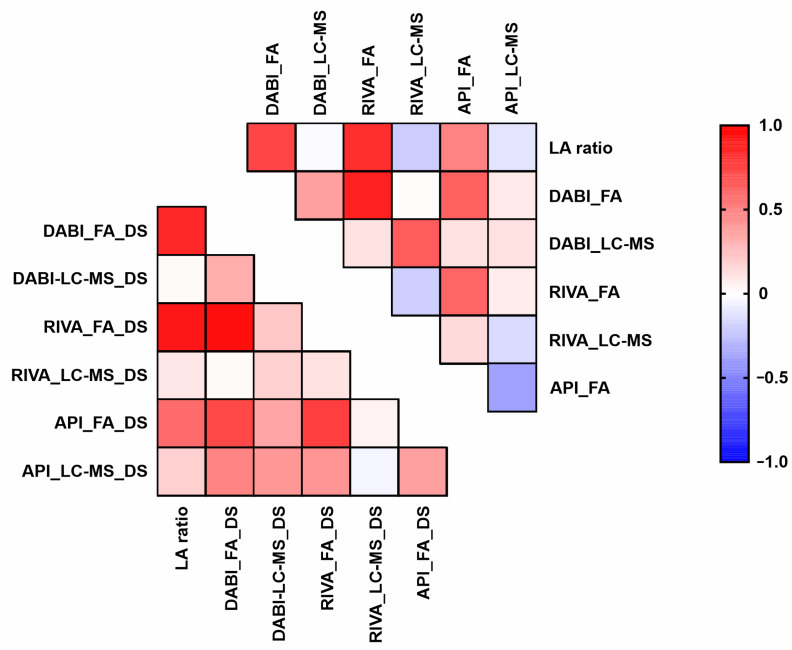
Heatmap of Spearman correlation coefficients of DABI, RIVA and API levels and LA ratio. Abbreviations: DS, DOAC-Stop procedure.

**Table 1 diagnostics-11-02027-t001:** Descriptive statistics and visualization of the data distribution of lupus anticoagulant (LA) ratio levels and concentrations of direct oral anticoagulant (DOAC) levels evaluated by the functional assays or by LC-MS method before and after DOAC-Stop procedure. All values are expressed in units ng/mL. Abbreviations: DABI, dabigatran; RIVA, rivaroxaban; API, apixaban.

Test	Median	Min	Max	Q1	Q3	Distribution
LA ratio	1.63	1.23	5.95	1.42	2.13	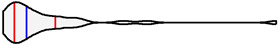
DABI, functional assay	96.9	68.0	235.0	76.9	114.8	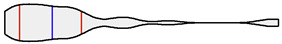
DABI, LC-MS	51.2	21.0	69.8	36.5	58.8	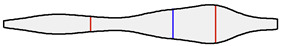
RIVA, functional assay	88.2	57.5	235.0	72.6	109.9	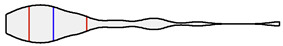
RIVA, LC-MS	83.0	31.9	107.8	49.9	92.8	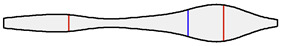
API, functional assay	106.1	70.8	235.0	91.1	115.1	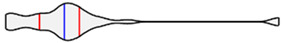
API, LC-MS	225.4	158.3	499.3	212.0	282.6	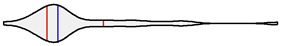
DABI, functional assay, DOAC-Stop	58.1	42.6	213.2	48.9	68.9	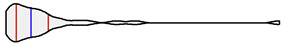
DABI, LC-MS, DOAC-Stop	0.28	0.10	2.56	0.19	0.33	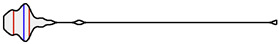
RIVA, functional assay, DOAC-Stop	55.5	43.5	218.3	48.8	77.9	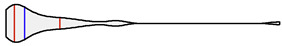
RIVA, LC-MS, DOAC-Stop	0.72	0.10	3.81	0.37	1.16	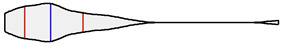
API, functional assay, DOAC-Stop	74.0	57.8	146.6	66.9	82.3	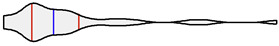
API, LC-MS, DOAC-Stop	3.60	0.10	34.74	2.13	6.68	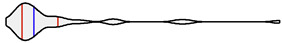

**Table 2 diagnostics-11-02027-t002:** Spearman correlation coefficient values and *p*-values of all tests with LA ratio. Statistically significant *p*-values are shown in bold.

Test	Correlation Coefficient	*p*-Value
DABI, functional assay	0.72	**<0.0001**
DABI, LC-MS	−0.02	0.9057
RIVA, functional assay	0.81	**<0.0001**
RIVA, LC-MS	−0.20	0.2787
API, functional assay	0.49	**0.0056**
API, LC-MS	−0.11	0.5627
DABI, functional assay, DOAC-Stop	0.84	**<0.0001**
DABI, LC-MS, DOAC-Stop	0.02	0.9218
RIVA, functional assay, DOAC-Stop	0.91	**<0.0001**
RIVA, LC-MS, DOAC-Stop	0.10	0.6207
API, functional assay, DOAC-Stop	0.58	**0.0006**
API, LC-MS, DOAC-Stop	0.18	0.3215

## Data Availability

All data generated in this study are included in this published article.

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
