# Peer review of "Evaluation of the Determination of Dabigatran, Rivaroxaban, and Apixaban in Lupus Anticoagulant-Positive Patients"

_diagnostics, 2021, doi:10.3390/diagnostics11112027_

Round 1
Reviewer 1 Report
In this paper, the authors evaluate a different aspect of the interface between DOACS and LA. Whereas most authors have evaluated the effect of DOACs on LA detection, the authors instead evaluated effect of LA on DOAC detection. As such, this is an interesting paper. However, there are a large number of mostly minor issues that detract from the reading. I have the following suggestions:
- The paper should be edited by a native English writer to increase the clarity of the text/message.
- There are a large number of grammatical and spelling errors throughout, which can easily be found by using Word’s spellcheck feature. This includes the title, where “at lupus anticoagulans” should read “in lupus anticoagulant”; similarly, in the abstract, “therapy at lupus” should read “therapy in lupus”; these minor issues continue throughout the manuscript; there are too many to identify here.
- The authors start their paper using ‘DOAC’ as the abbreviation, and about half way through change this to NOAC and NOACs for the rest of the paper, for unexplained reason. Keep consistent throughout and use DOAC.
- Similarly, the authors use several different terms for LA throughout; examples: “anti-lupus anticoagulants”, “lupus anticoagulants inhibitor”, “lupus anticoagulant type inhibitor”, “lupus LSR”, “Lupus anticoagulants (LSR)”, “lac screen ration (LSR)” ; Just call it lupus anticoagulant and just abbreviate it as LA (or LAC if preferred). For ratio, just use “LA ratio”. Also, define on first appearance, not on page 3 where it appears.
- Abstract: “with DOAC develop LA of medium and higher titers, we do not recommend checking with functional tests” suggest ““with DOAC develop LA of medium and higher titers, we do not recommend checking DOAC levels with functional tests”. Since LA testing is a form of functional test and readers may get confused.
- Authors use “dabigatran etexilate” in the beginning, which is the prodrug; are they measuring this or ‘dabigatran’ (the active drug) – be clear throughout, and when talking about results by functional assays and by LC-MS.
- Page 2: “recommended to treat DOAC in patients with lupus anticoagulants because studies have not shown interference with VKA,” has unclear meaning
- Page 2: “precipitation” is an unusual word to use here; change to ‘coagulation’
- Page 2: last paragraph on rivaroxaban. Why is additional text on apixaban not included here?
- Page 2: “on basis recommendation of ISTH” add reference
- Page 2: what blood volume was collected into what size citrate tubes? Only 0.5mL plasma collected from top infers a 1mL tube, but that would be impossible for the experiments described.
- Page 3: “Dabigatran, rivaroxaban, and its deuterated analog DAB-D3” is DAB-D3 a deuterated analog of rivaroxaban?
- Page 3: “Lupus anticoagulants (LA) tests are performed by LAC Screen tests (Werfen, Barcelona, Spain).” Hopefully also paired with a confirm reagent and reported as a normalized ratio. This section needs major revision to clarify what the authors are measuring here as LA and as LA ratio.
- Page 5: “as 1.9, 1,1, and 0,5 for DABI, RIVA, API,” mixed use of dot and comma separators for decimal points; DABI, RIVA, API not yet defined and why use these abbreviations in the text?
- Several versions also of “Anti-Xa” “antiXa” “anti Xa”
- Page 6: “dicoumarins”? do you mean Vitamin K antagonists? “gatrans and xabans”? only one gatran, so no point calling it this; xaban never really defined, and I don’t think it helps calling it this here.
- Page 6: “lipemia, chylosis and haemolysis” chylosis not often used; perhaps bilirubin?
Author Response
Reviewer 1:
We would like to gratefully thank the reviewer for his detail review and we would like to apologize for the many stylistic errors in the text, which we have tried to eliminate as much as possible in this revision.
1. The paper should be edited by a native English writer to increase the clarity of the text/message.
- The text was fully revised by native English speaker and processed by Writefull software.
2. There are a large number of grammatical and spelling errors throughout, which can easily be found by using Word’s spellcheck feature. This includes the title, where “at lupus anticoagulans” should read “in lupus anticoagulant”; similarly, in the abstract, “therapy at lupus” should read “therapy in lupus”; these minor issues continue throughout the manuscript; there are too many to identify here.
- The text was fully revised by native english speaker and processed by Writefull software.
3. The authors start their paper using ‘DOAC’ as the abbreviation, and about half way through change this to NOAC and NOACs for the rest of the paper, for unexplained reason. Keep consistent throughout and use DOAC.
- We corrected abbreviation of NOAC to DOAC.
4. Similarly, the authors use several different terms for LA throughout; examples: “anti-lupus anticoagulants”, “lupus anticoagulants inhibitor”, “lupus anticoagulant type inhibitor”, “lupus LSR”, “Lupus anticoagulants (LSR)”, “lac screen ration (LSR)” ; Just call it lupus anticoagulant and just abbreviate it as LA (or LAC if preferred). For ratio, just use “LA ratio”. Also, define on first appearance, not on page 3 where it appears.
- We corrected and unified all terms to abbreviation of LA.
5. Abstract: “with DOAC develop LA of medium and higher titers, we do not recommend checking with functional tests” suggest ““with DOAC develop LA of medium and higher titers, we do not recommend checking DOAC levels with functional tests”. Since LA testing is a form of functional test and readers may get confused.
- We corrected sentence as suggested.
6. Authors use “dabigatran etexilate” in the beginning, which is the prodrug; are they measuring this or ‘dabigatran’ (the active drug) – be clear throughout, and when talking about results by functional assays and by LC-MS.
- We changed the etexilate dabigatran into dabigatran in the abstract as suggested.
7. Page 2: “recommended to treat DOAC in patients with lupus anticoagulants because studies have not shown interference with VKA,” has unclear meaning
- Moreover, although it is not currently recommended to treat LA patients with DOAC as studies have not shown interference with vitamin K antagonists, many patients with LA are, in fact treated with DOAC.
8. Page 2: “precipitation” is an unusual word to use here; change to ‘coagulation’
- We changed the precipitation into coagulation in the abstract the as suggested.
9. Page 2: last paragraph on rivaroxaban. Why is additional text on apixaban not included here?
- We updated the paragraph as shown below including information about apixaban accordingly:
“Anti-factor Xa chromogenic assays can accurately measure a wide range of plasma concentrations provided that a standard calibration curve is established with calibrators and controls of rivaroxaban or apixaban calibrators and controls [5,6]. A study of 23 centers showed that chromogenic assays against factor Xa in conjunction with rivaroxaban calibrators and controls can measure plasma concentrations of rivaroxaban in the range of 20–660 μg/l, but limited data are described for apixaban [6]. In addition, the mean measured concentrations of rivaroxaban were in agreement with the expected values, even at low rivaroxaban concentrations, when a modified STA® Rotachrom® test (Diagnostica Stago) was used.”
10. Page 2: “on basis recommendation of ISTH” add reference
- The reference was added.
11. Page 2: what blood volume was collected into what size citrate tubes? Only 0.5mL plasma collected from top infers a 1mL tube, but that would be impossible for the experiments described.
- Blood volume was added into sentence.
12. Page 3: “Dabigatran, rivaroxaban, and its deuterated analog DAB-D3” is DAB-D3 a deuterated analog of rivaroxaban?
- We corrected the sentence accordingly:
“Dabigatran, rivaroxaban, and stable isotopically labelled D3-dabigatran (Toronto Research Chemicals Inc, Toronto, Canada) and apixaban (Pfizer Inc, USA) were dissolved in methanol (LC-MS quality, Sigma, USA) at a final concentration of 1 mg/ml expressed as free substances.”
13. Page 3: “Lupus anticoagulants (LA) tests are performed by LAC Screen tests (Werfen, Barcelona, Spain).” Hopefully also paired with a confirm reagent and reported as a normalized ratio. This section needs major revision to clarify what the authors are measuring here as LA and as LA ratio.
- We corrected the paragraph accordingly:
“LA tests are performed using LAC Screen tests (Werfen, Barcelona, Spain) with improved specificity based on the diluted Russell’s viper venom tests (dRVVT). The test detects the group of antiphospholipid antibodies that are directed against negatively charged phospholipids or complexes between phospholipids and proteins (either beta-2-glycoprotein 1 or clotting factors such as prothrombin) and that prolonged coagulation tests. The dRVVT test is performed as a screen (low phospholipid concentration) and confirm (high phospholipid concentration) and is always expressed as a ratio to normal plasma. Pathology is defined by a ratio greater than 1.2.”
14. Page 5: “as 1.9, 1,1, and 0,5 for DABI, RIVA, API,” mixed use of dot and comma separators for decimal points; DABI, RIVA, API not yet defined and why use these abbreviations in the text?
- We corrected dots and unified abbreviations for the mentioned drugs in the entire text.
15. Several versions also of “Anti-Xa” “antiXa” “anti Xa”
- We unified abbreviations into Anti-Xa in the entire text.
16. Page 6: “dicoumarins”? do you mean Vitamin K antagonists? “gatrans and xabans”? only one gatran, so no point calling it this; xaban never really defined, and I don’t think it helps calling it this here.
- We corrected the sentences accordingly:
“The new oral anticoagulants are revolutionary drugs in the field of anticoagulation and show clearly better results in antithrombotic prevention after the era of Vitamin K antagonists.“
“For this reason, specific functional assays have been developed for dabigatran, rivaroxaban and apixaban based on diluted thrombin time [17] and the anti-Xa activity, respectively.”
17. Page 6: “lipemia, chylosis and haemolysis” chylosis not often used; perhaps bilirubin?
- We corrected the sentence.

Reviewer 2 Report
1. The authors should clarify whether the study is prospective or ambispective.
2. In addition, the registration number of the Clinical Research Ethics Committee that evaluated the study should be indicated.
3. The discussion should indicate how LA interferes with the measurement of coumarin and heparin with respect to DOACs.
Author Response
Reviewer 2
We would like to thank the reviewer for suggestions.
- The authors should clarify whether the study is prospective or ambispective.
- This was a retrospective study. This information has been added into Abstract and Methods.
In addition, the registration number of the Clinical Research Ethics Committee that evaluated the study should be indicated.
- The following sentence was updated in the Institutional Review Board Statement as follows:
“The study was conducted according to the guidelines of the Declaration of Helsinki, according with the Institutional Ethics Committee of University Hospital Olomouc (approval number: 39/21 and form number: Fm-L009-001-HOK-026).”
The discussion should indicate how LA interferes with the measurement of coumarin and heparin with respect to DOACs.
- These sentences have been added into the Discussion:
“However, we are aware that these specific tests may be affected by the administration of heparin. On the one hand, unfractionated heparin, whose main target is thrombin, affects dTT, whereas or low molecular weight heparin, whose main target is Factor-Xa, affects the specific determination of the activity of this factor.”

Round 2
Reviewer 1 Report
I thank the authors for their revision, which has made the paper more readable and improved clarity. The tracked changed manuscript is, however, difficult to read; there appears to be some minor residual issues:
Page 2: Lines 47-51: sentence starting “Despite the contraindication, …” needs revision, including the specific use of the word ‘who’; it doesn’t seem to make sense. Ditto page 2, line 61 onwards; ‘who’ relates to people, not events; perhaps use ‘that’?
Author Response
We would like to thank the reviewer for the review. At the reviewer's suggestion, we have corrected lines 43-49 in the Introduction as shown below:
“Although warfarin is still the preferred treatment for patients with LA (except for APS syndrome) due to its extensive clinical experience, many patients are currently treated primarily with DOACs in practice. Furthermore, patients may also develop LA secondary to the course of treatment for atrial fibrillation, prevention of ischemic stroke and thromboembolic events. These patients, in whom DOAC levels need to be monitored in some situations, are the focus of our study.”